# Comparison of Tomato Transcriptomic Profiles Reveals Overlapping Patterns in Abiotic and Biotic Stress Responses

**DOI:** 10.3390/ijms24044061

**Published:** 2023-02-17

**Authors:** Ciro Gianmaria Amoroso, Daniela D’Esposito, Riccardo Aiese Cigliano, Maria Raffaella Ercolano

**Affiliations:** 1Department of Agricultural Science, University of Naples “Federico II,” 80055 Portici, Italy; 2Sequentia Biotech SL, Calle Comte D’Urgell 240, 08036 Barcelona, Spain

**Keywords:** plant stress, transcriptomics, transcription factors, signaling, cell wall, phytohormones, genome editing

## Abstract

Until a few years ago, many studies focused on the transcriptomic response to single stresses. However, tomato cultivations are often constrained by a wide range of biotic and abiotic stress that can occur singularly or in combination, and several genes can be involved in the defensive mechanism response. Therefore, we analyzed and compared the transcriptomic responses of resistant and susceptible genotypes to seven biotic stresses (*Cladosporium fulvum*, *Phytophthora infestans*, *Pseudomonas syringae*, *Ralstonia solanacearum*, *Sclerotinia sclerotiorum*, Tomato spotted wilt virus (TSWV) and *Tuta absoluta*) and five abiotic stresses (drought, salinity, low temperatures, and oxidative stress) to identify genes involved in response to multiple stressors. With this approach, we found genes encoding for TFs, phytohormones, or participating in signaling and cell wall metabolic processes, participating in defense against various biotic and abiotic stress. Moreover, a total of 1474 DEGs were commonly found between biotic and abiotic stress. Among these, 67 DEGs were involved in response to at least four different stresses. In particular, we found RLKs, MAPKs, Fasciclin-like arabinogalactans (FLAs), glycosyltransferases, genes involved in the auxin, ET, and JA pathways, MYBs, bZIPs, WRKYs and ERFs genes. Detected genes responsive to multiple stress might be further investigated with biotechnological approaches to effectively improve plant tolerance in the field.

## 1. Introduction

Plants are sessile living organisms that developed many strategies for quickly adapting to environmental changes. Despite this, adverse environmental factors (abiotic stress), such as drought, salinity, low temperatures, oxidative stress and plant pathogen or pest attacks (biotic stress), can negatively affect plant growth and production [1]. Simultaneous exposition to biotic and abiotic stress can induce tremendous crop yield losses. Therefore, the resistance mechanisms to various tomato stresses have been under investigation for a long time. Until a few years ago, most stress-related studies focused on the single stress response mechanism. Recently, more emphasis has been given to studies investigating the plant response to combinations of multiple stresses [2]. An increasing number of studies have been conducted to identify new forms of resistance to multiple stressors, and several genes that recognize both biotic and abiotic stress have been found [3,4,5,6,7]. In tomato, different genes for perception, signaling, hormone balancing, and transcription modulation are involved in various biotic and abiotic stress responses [8,9,10,11,12]. The modulation of these pathways leads to the activation or repression of several responsive proteins. In this context, genetic engineering became a fundamental tool for creating new plants that quickly adapt to biotic and abiotic stress without compromising plants’ phenotypical traits and yields [1,13]. For this purpose, studying transcriptomic alterations in plants subjected to various stress could aid the identification of principal genes that participate in resistance or susceptibility processes [14]. High-throughput sequencing RNA-seq technology quantifies gene expression levels with high accuracy [15]. To date, RNA-seq has been extensively used to investigate plant stress interactions, and raw reads of each experiment are deposited in sequence databases. Therefore, published sequencing data could represent a valuable resource to further explore plant stress responses by analyzing and comparing different studies to identify genes responsive to various stresses.

In tomato, a comparative analysis using the microarray gene expression technique identified 1862 and 835 genes responding to biotic and abiotic stress, respectively [16]. However, RNA-seq technology showed a higher sensitivity for gene expression than microarray technology and better genome coverage [17]. So far, Illumina technology is the most used sequencing platform for RNA samples due to its accuracy, rapidity, and moderate price [18]. In our work, RNA-seq raw datasets of tomato-stressed samples were assessed to identify genes involved in both biotic and abiotic stress. Raw data included tomato transcriptional response to 12 different stressors, eight biotic collected by [19], and four abiotic stress collected from other works [20,21,22]. In particular, genes involved in cell wall metabolism, membrane receptors, transcription factors (TFs), and phytohormones modulation were deeply investigated. The final goal of our analysis was to prioritize a list of responsive genes to multiple stress as potential candidates to be employed in genetic engineering programs. Indeed, genes resulting from our comparative analysis could be further characterized through biotechnological approaches to investigate their role in tomato response to multiple stresses.

## 2. Results

This work aimed to explore the transcriptomic alterations of tomato plants exposed to pathogens and environmental stresses to identify genes involved in response to multiple stressors. To examine and characterize genes responding to different stresses, we exhaustively re-analyzed twelve publicly available RNA-seq studies of resistant (R) and susceptible (S) tomatoes challenged by different biotic and abiotic stresses (Table 1, Appendix A). 

The experiments were singularly analyzed, and the lists of resulting DEGs were compared to identify genes involved in response to different stresses (Appendix A). A high variable number of DEGs was found among studies depending on the induced stress (Table 1). *P. infestans*, *P. syringae*, *S. sclerotiorum*, and *T. absoluta* (T genotype) caused the differential expression (DE) of a considerable number of tomato genes (higher than 10,000). TSWV-stressed plants showed a DEGs peak at twenty-one dpi (1490 DEGs), while R. solanacearum induced an increased number of DEGs at two dpi. The R and S genotypes to *C. fulvum* showed a decrease in DEGs from seven to twenty dpi. Among abiotic stress analyzed, DEGs ranged from 2000 to 6000, except for the low temperatures experiment (12836 DEGs).

To have an overview of DEGs responsive to different stresses, we first analyzed biotic and abiotic stress separately, seeking DEGs in more than a single experiment. Then, we extended the comparison of the results for detecting genes involved in both biotic and abiotic responses.

### 2.1. Exploration of the Datasets of DEGs under Various Biotic Stress

Transcriptomic datasets of plants subjected to biotic stress were gathered in four broad groups: (1) fungi, including *C. fulvum*, *P. infestans* (R and S response) and *S. sclerotiorum* (S response); (2) bacteria, containing *R. solanacearum* (R and S response) and *P. Syringae* (S response); (3) TSWV (R response), and (4) *T. absoluta* (T and Sresponse). To better analyze the data within each biotic stress group, we considered the more similar time points. Therefore, tomato response to fungi, TSWV and *T. absoluta* was investigated at more than 20 dpi, whereas transcriptomic changes inplants infected with bacteria were investigated at 2 dpi. Appendix A shows the DEGs induced or repressed by each tomato interaction studied. 

The DEGs found in each group (fungi, bacteria, TSWV, and *T. absoluta*) were intersected to identify R and S common genes. The S genotypes to fungi showed 913 down and 560 common-upregulated DEGs (Figure 1A), while R plants showed an overlap of 1841 and 1770 up and downregulated genes, respectively (Figure 1B). The S tomatoes to bacteria shared 755 up and 732 downregulated genes (Figure 1C) while the R genotype showed 1247 down and 1135 upregulated genes (Figure 1D). Considering that in this study TSWV and *T. absoluta* could not be compared with other viruses or pests, we used their whole datasets of DEGs for further analysis.

Contrasting expression patterns among interactions were also found. For example, in R genotypes to fungi, 298 genes were downregulated in *C. fulvum* and upregulated in *P. infestans*. On the contrary, 592 genes were upregulated in *C. fulvum* and downregulated in *P. infestans* (Figure 1). To identify genes associated with resistance or susceptibility within each pathogen group, lists of common DEGs identified in R and S genotypes were compared (Figure 2). The R genotypes to fungi specifically induced and repressed 1457 and 1090 genes, respectively (Figure 2), while S genotypes showed 176 and 233 privately up and downregulated genes. Similarly, the R genotypes to bacteria showed the private activation of 826 genes, while 989 were downregulated. The S genotypes displayed 446 and 474 specific up and downregulated genes (Figure 2). The R genotype to *T. absoluta* showed the private differential regulation of 2432 induced and 2773 repressed genes, while the S genotype showed 536 upregulated and 419 downregulated private genes (Figure 2). Lists of private DEGs were used to analyze processes related to biotic stress response in R and S genotypes.

#### 2.1.1. Analysis of Processes Putatively Involved in Biotic Stress Response, Privately Activated by Resistant and Susceptible Genotypes

The R genotypes to fungi showed the activation of genes involved in ethylene (ET) and jasmonic acid (JA) pathways, while several genes for the synthesis of abscisic acid (ABA) were repressed. By contrast, S plants showed the repression of the ET pathway, while ABA metabolism was generally activated. In addition, R and S genotypes differed in regulating genes participating in cell wall metabolism, in signaling or encoding for transcription factors (TFs), such as WRKYs and MYBs, and pathogen-related proteins (PR-proteins) which were mainly upregulated in R genotypes (Figure 3). 

The R genotype to bacteria showed the induction of genes for salicylic acid (SA) synthesis, while ABA and JA synthesis were predominantly repressed. In addition, the R genotype to bacteria showed the downregulation of different TFs, especially ERFs, DOFs, and MYBs. On the other hand, different MYBs were upregulated in S genotypes (Figure 3). 

The T genotype to *T. absoluta* upregulated genes involved in lignin, AGPs, and hemicellulose biosynthesis, simultaneously activating SA synthesis. By contrast, we observed the repression of genes involved in ET and ABA biosynthesis. Moreover, several genes participating in signaling and proteolysis processes were DE. Contrarily, the S genotype mainly upregulated the ET pathway (Figure 3).

The R genotype to TSWV showed general repression of the defense signaling and the activation of cell wall metabolism (Figure 3).

#### 2.1.2. Differentially Regulation of Cell Wall Precursors under Various Biotic Stress

Due to the cell wall’s role in different stress responses, we focused on genes participating in the biosynthesis of the cell wall precursors, with specific reference to genes participating in cellulose and pectin metabolism such as Fasciclin-like arabinogalactans (FLAs), expansins (EXPs), pectate lyases, UDP-glucose-4-epimerases (UGEs) and polygalacturonases (PGs) (Figure 4). 

Cellulose is one of the main components of the cell wall. In this study, two FLAs (Solyc01g091530 and Solyc10g005960) were downregulated among R genotypes to fungi, while Solyc07g045440 and Solyc07g053540 were also repressed by bacteria S and induced by TSWV R. The Solyc06g075220 (FLA8) was downregulated by S genotypes to bacteria and upregulated by the T genotype to *T. absoluta*. By contrast, Solyc12g015690 (FLA11) was upregulated by both *T. absoluta* T and TSWV R genotypes. Cellulose disassembly could also be promoted by EXP genes. Interestingly, an Expansin-like protein, Solyc08g077900 (EXLB1), was upregulated by R genotypes to fungi. 

Among genes involved in pectin degradation, we found that pectate lyases were involved in response to different pathogens. For example, the R genotypes to fungi and *T. absoluta* activated Solyc03g111690, while R genotypes to TSWV and *T. absoluta* commonly induced Solyc05g014000 and Solyc06g083580. Similarly, two pectate lyases (Solyc09g008380 and Solyc09g091430) were activated by the R genotype to TSWV and the T genotype to *T. absoluta* and repressed by S genotypes to bacteria.

Other important enzymes involved in pectin metabolism were found DE under various biotic stress. In particular, a PG (Solyc08g060970) and a UGE (Solyc02g030230) were induced by R genotypes to fungi and bacteria. Similarly, the R genotypes to bacteria showed the downregulation of Solyc12g010540. By contrast, a different UGE (Solyc07g043550) was activated by bacteria S genotypes and repressed by the *T. absoluta* T genotype.

Galacturonic acid (GalA) is utilized to synthesize hemicellulose and pectin. Our results showed that T plants to *T. absoluta* activated four genes participating in the GalA epimerization (GAE) (Solyc01g091200, Solyc09g092330, Solyc05g050990, and Solyc10g018260), while Solyc08g079440 and Solyc08g082440, involved in UDP-galactose production, were repressed (Figure 4).

In general, we observed that R and T genotypes to fungi and *T. absoluta* showed a high number of common DEGs. Among these, we found the activation of an endo-1,4 beta-glucanase (Solyc05g005080), an endo-1,4-mannosidase (Solyc10g074920), and a 3,5-epimerase/4-reductase (Solyc08g080140).

#### 2.1.3. Identification of Genes Differentially Expressed under Different Biotic Stress and Involved in the Signaling Process

The investigation of DEGS involved in the plant signaling process allowed the identification of 30 genes encoding for RLPs (Figure 5). In particular, the analysis of the seven expression profiles pointed out a clear divergent regulation. Interestingly, an LRR-RLK (Solyc08g061560) was suppressed by R genotypes to fungi and bacteria and induced by *T. absoluta* T and TSWV R genotypes. Similarly, Solyc03g093460 was activated by the T genotype to *T. absoluta* (Figure 5) and repressed by R genotypes to fungi and bacteria. We also found two common RLKs (Solyc02g068830 and Solyc06g048740) differentially regulated among R genotypes to fungi, bacteria, and TSWV, while Solyc11g006040 was induced in R genotypes to fungi, *T. absoluta* and TSWV (Figure 5).

### 2.2. Exploration of the Datasets of Differentially Expressed Genes under Various Abiotic Stress

Lists of DEGs resulting from tomatoes stressed with drought, low temperature, salinity, and oxidative stress were compared to identify genes involved in response to various abiotic stress. The two studies on drought stress showed similar expression profiles except for a few genes (Figure 6A). Therefore, to select common DEGs in the susceptible responses to drought, we compared DEGs of M82 and Jinlingmeiyu (S cultivars), identifying 176 up and 347 downregulated genes (Figure 6B). Moreover, we compared DEGs of S genotypes with those of the IL9-1 (T plants), discovering 28 up and 32 downregulated genes privately DE by the T genotype (Figure 6B).

#### 2.2.1. Investigation of Hormones and TFs Differentially Regulated under Various Abiotic Stress

To identify genes responsive to multiple abiotic stress, we focused on four primary defensive processes: signaling, hormone modulation, TFs regulation and biosynthesis of cell wall precursors. Our results showed that the salinity induced DEGs implicated in ABA, ET, and SA pathways. By contrast, low temperatures caused the downregulation of genes involved in SA, auxins, and brassinosteroids (BRs) pathways (Figure 7).

Among the 32 private DEGs found in the drought T genotype, Solyc01g110680 and Solyc12g096820 were, respectively, involved in the auxin and SA biosynthesis and were suppressed. Contrarily, the S genotypes revealed the activation of a gene involved in ABA signaling (*HVA22d*) and the repression of genes involved in ET pathways (Solyc02g064950, Solyc07g049550, Solyc08g079750) (Figure 7).

Differences in TFs regulation included ERFs, which were mainly induced during salinity and oxidative stresses, while Solyc07g054220 (*ERF2a*) was also upregulated by the drought T genotype. In addition, the drought T genotype showed the activation of various NAC genes, including Solyc01g009860. Different WRKYs were upregulated during salinity, low temperature, oxidative stress, and in S genotypes to drought. On the other hand, the T genotype to drought showed the downregulation of Solyc07g055280 (*WRKY78*).

#### 2.2.2. Investigating Genes Involved in Different Abiotic Stress Responses and Participating in Signaling and Cell Wall Processe

Comparison of genes involved in the signaling process led to the identification of genes encoding for Calnexins, RLKs, and Glutamate receptors (GLRs) (Table 2). Interestingly, Solyc03g118040 (Calnexin) was exclusively downregulated by the T genotype to drought and upregulated during all the other stress. At the same time, two RLKs (Solyc02g072310 and Solyc05g056370) were repressed during low temperature, salinity, and oxidative stress. Genes encoding for Glutamate receptors were mainly downregulated under different abiotic stress. In particular, Solyc06g063180 and Solyc07g052390 were repressed by salinity and low temperatures. The Solyc07g052400 was downregulated during drought S, oxidation, and low temperatures, while low temperatures and oxidation repressed Solyc05g045650. By contrast, Solyc02g067030 (*SNF1*) was induced by salinity and downregulated by low temperatures (Table 2). 

The cell wall precursors showed common DE of glucose 6-dehydrogenases (UGDs), glucuronate 4-epimerases (GAEs), glucose-4-epimerases (UGEs) and cellulose synthase-like (CSL) genes, repressed during salinity and oxidative stress and induced by low temperatures (Table 2). Two GAEs showed a divergent expression pattern during salinity, low temperature, and drought stress (Solyc07g006220 and Solyc05g050990), while the drought T genotype repressed a glycosyltransferase (Solyc12g096830). Finally, a CSL (Solyc03g097050) was induced by all the abiotic stress analyzed and was exclusively repressed by the drought T genotype.

### 2.3. Identification of Common Genes Differentially Expressed under Biotic and Abiotic Stress

Comparing lists of private DEGs obtained for each group of biotic (fungi, bacteria, TSWV and *T. absoluta*) and abiotic stress (drought, low temperature, salinity, and oxidative stress), we identified common genes related to signaling, cell wall, hormones, and TFs, activated or repressed under different stresses. Generally, the R genotypes to pathogens showed a higher number of common DEGs with abiotic stresses than their S counterparts for most of the gene classes (Figure 8). An exception was the identification in R and S plants to bacteria of a similar number of common DEGs, involved in the signaling and cell wall process, with low temperature stress. Due to the low number of private DEGs in T and S genotypes to drought, few genes were identified in common with biotic stress (Figure 8) and have been reported in Appendix A.

A total of 1474 common DEGs were found between biotic and abiotic stress (Appendix A). Therefore, we focused on those genes simultaneously DE in at least one abiotic stress and three groups of biotic stresses (Figure 9). Ten genes were involved in the biosynthesis of cell wall compounds, thirty-five DEGs resulted involved in signaling, thirteen DEGs encoded for hormones, and nine DEGs were annotated as TFs (Figure 9). DEGs involved in the synthesis of cell wall compounds included FLAs, glycosyltransferases (GTs), beta-xylosidase, GAEs, polygalacturonases, pectate lyases, and peptidoglycan-binding LysM domain-containing proteins (Figure 9).

Genes belonging to the signaling process included RLKs, mitogen-activated protein kinases (MAPKs), a pathogenesis-related (PAR) gene involved in calcium signaling, a phosphoinositide phospholipase C, a Rop-interactive crib motif-containing protein, and genes involved in the photosynthetic pathway (Figure 9). Interestingly, we also found the upregulation of the *Ve2* gene (Solyc09g005080) in the *C. fulvum* R genotype, *S. sclerotiorum*, *P. syringae*, oxidative stress, and its downregulation in the TSWV R genotype.

Hormone regulation showed various genes involved in the auxin and ABA pathways mainly activated by R genotypes to fungi and abiotic stress, while ET and JA synthesis showed a divergent expression profile during various biotic and abiotic stress (Figure 9). TFs included genes encoding for MYBs, WRKYs, ERFs, bZIPs, GRAS, and AP2/ERF (Figure 9).

## 3. Discussion

Using a common pipeline to analyze and compare publicly RNA-seq studies of tomato plants exposed to different stresses, we obtained an overview of the pathways involved in cellular reprogramming under different stresses. Moreover, we shed light on genes differentially expressed during various tomato stress interactions. The comparative analysis of tomato transcriptomic profiles provided a better understanding of the complex mechanisms underpinning the plant defense process. 

### 3.1. Biotic Stressors Induced a Variegated Transcriptomic Response in Tomato

Comparative analysis of eight transcriptomics experiments, including resistant (R) and susceptible (S) genotypes to fungi, bacteria, *T. absoluta* and TSWV, allowed us to look into the intricate process of the tomato response to biotic stress. Plant-pathogen interaction leads to a deep remodeling of transcriptomic and metabolic pathways, such as phytohormones, signaling proteins, TFs, and cell wall-related processes [1,3,30,31]. 

Plant hormones are wellknown as critical regulators of signal defense responses in plants. In this work, we focused on the process involved in basal resistance mechanisms. Our findings suggested that the simultaneous activation of JA and the repression of ABA signaling could be a key factor in activating resistant basal response to fungi [32]. However, it is important to mention that different fine-tuned hormonal regulations could be observed depending on the pathogens’ lifestyle [33,34]. In contrast with fungi, the repression of JA and the activation of SA promoted the immune response in the R genotype to *R. solanacerarum*, confirming previous findings [35,36]. Differences in hormone regulation may also induce plants’ anti-herbivore characteristics [37]. In particular, we found that several genes involved in SA metabolism were induced by the T genotype to *T. absoluta* and could be involved in the tolerance process [38].

The cell wall represents the first barrier for pathogen perception and the activation of defensive responses against multiple biotic stressors immediately downstream of the cuticle layer. Hence, changes in cell wall composition and structure may lead to specific signaling cascades and pathogen responses [39]. An interesting subclass of arabinogalactan proteins (AGPs) involved in cellulose metabolism, named FLAs, was highly challenged in our samples. Such proteins can have multiple roles in plant signaling, growth, development and stress responses [40,41,42] and may regulate lignin and cellulose synthesis/deposition in response to mechanical stimuli [43].

In our study, FLAs were particularly induced by fungi R genotypes. For example, Solyc07g045440 (*FLA2*) was upregulated by fungi R and *T. absoluta* T, while Solyc01g091530 (*FLA13*) was activated by fungi R and TSWV R genotypes. A recent study reported that in *Nicotiana benthamiana* different FLAs were selectively repressed after the infection with turnip mosaic virus (TuMV) and *Pseudomonas syringae* pv tomato strain DC3000 (Pst DC3000) [44]. These genes were also identified as induced during *M. incognita* infections and downregulated during the combination of water stress and nematodes [45]. Other important cell wall components, such as hemicellulose and pectin, were affected by multiple biotic stress. Interestingly, the R genotype to bacteria showed the downregulation of a GDP-mannose-4,6-dehydratase (Solyc12g010540), involved in the pectin metabolism [46]. Moreover, R genotypes to fungi and bacteria showed the common activation of a UGE (Solyc02g030230) and a PG (Solyc08g060970) that could have a role in the resistance against pathogens and abiotic stress [47,48,49]. Pathogens often induce enzymes implicated in pectin degradation, such as pectate lyases, to promote plant infections [50]. Here, we found that two pectate lyases were induced by the *T. absoluta* T genotype and repressed by the R genotype to bacteria. Hence, we speculate that their downregulation may be important for maintaining proper cell wall properties during bacterial infections. In our work, several pectate lyases were responsive to *T. absoluta* and TSWV. For instance, Solyc06g083580 and Solyc05g014000, induced by both T and R genotypes, have been found in other studies as involved in mites and potato *P. infestans* susceptibility [51,52]. It is worth noting that pectate lyases may have a key role in biotic stress response since their silencing can provide resistance against different pathogens [53].

Receptors and signaling molecules play a crucial role in the capability of plants to respond adequately to specific stresses [7]. Our study found RLKs with divergent expression patterns during various stress. For example, Solyc06g048740 was induced by R genotypes to fungi and bacteria and repressed by R plants to TSWV. This gene was also upregulated in a tomato *C. fulvum* R genotype [54]. Moreover, Solyc07g006480 was exclusively activated by R plants to bacteria and repressed by fungi and TSWV R plants. This gene was also activated in salinity-tolerant plant roots [55]. Our results suggest that RLKs may play essential roles in fine-tuning the signaling process and their expression pattern can promote specific defense responses against different stressors. The Solyc04g014400, a *Pseudomonas*-responsive RLP gene [56], also showed a common downregulation in all R genotypes, as well as Solyc11g056680 that was induced by potato cyst nematode (PCN) in tomato roots [57], and could be involved in response to multiple pathogens.

### 3.2. Tomato Transcriptomic Reprogramming under Different Abiotic Stresses

This work analyzed tomato response to four different abiotic stress (drought, salinity, low temperature and oxidative stress). Adaptive plant responses to specific abiotic stresses are fine-tuned by a network of hormonal signaling cascades, including ABA, ET, JA, and SA [58,59]. In our work, the auxins pathway was mainly repressed during drought stress, accordingly to [60]. We also found that the R genotype to drought downregulated the Solyc12g096830, orthologs of AT1G05680 (*UGT74E2*) involved in auxin distribution and drought stress response [61]. By contrast, the susceptible genotypes showed the induction of Solyc11g010930, which encodes for a drought-responsive protein also involved in salinity stress response [62]. Furthermore, S genotypes to drought showed the repression of genes involved in ET biosyntheses such as Solyc07g049550 (ACCO) and Solyc08g079750 (ACC synthase). The crosstalk between ABA and ET synthesis is important for the proper growth in drought conditions and for the closure and opening of stomata cells under drought and salinity stress [63]. In addition, we found several genes involved in SA biosynthesis induced under salinity to mitigate the deleterious effect of the stress, as reported in [64]. Despite this, the excessive production of SA could decrease plant tolerance [65]. On the contrary, plants subjected to low temperatures downregulated genes for SA, which instead could enhance plant growth and production [65,66]. 

TFs are fundamental tools to coordinate the transduction of various stress signals in plants during abiotic stresses. In this work, we found the common upregulation of Solyc07g054220 (*ERF2a*) during salt, oxidative, and drought T responses. This gene is known to exhibit different expression patterns under various abiotic stress and could be crucial for governing multiple abiotic stress interactions [67]. Moreover, the drought T genotype showed the private upregulation of a NAC gene (Solyc01g009860) orthologs of AT5G13180 (*ANAC083*) involved in leaf senescence, ABA and drought stress responses, negatively regulating the xylem vessels formation [68,69].

Changes in Ca^2+^ concentration led to the activation of plant signaling. Interestingly, a calnexin (Solyc03g118040) was induced during salinity, low temperatures and oxidative stresses and exclusively repressed by the T genotype to drought. This gene is involved in calcium homeostasis by binding the Ca^2+^ and participates in plant adaption to unfavorable environmental conditions [70,71]. This study identified four GLRs (Solyc06g063180, Solyc07g052390, Solyc07g052400, and Solyc05g045650) reported as putative PM-Localized Proteins related to Ca^2+^ transport that may induce a specific spectrum of downstream responses for fine-tuning adaptive responses to abiotic stress [72]. Among these, Solyc07g052400 was also found localized in a QTL region for tolerance to water deficit in tomato [73]. 

Salinity and oxidative stress induced the upregulation of four genes encoding for AXS, UGD, and GAE proteins that were downregulated during low temperatures. The *SNF1* gene showed an opposite regulation during salinity and low temperatures and was identified as an important factor in response to drought and cold stress by interacting with a second gene (*ShCIGT*), promoting plant resistance against multiple abiotic stress [74]. Interestingly, we also found a cellulose synthase-like (CSL) gene (Solyc03g097050) that was DE during all the abiotic stress and showed an opposite regulation between the T and the S genotypes to drought. Cellulose synthase genes might enhance plant tolerance against drought and oxidative stress and are involved in ABA modulation during abiotic stress [75,76,77]. A recent study suggested that this gene could also be related to tomato yellow leaf curl virus (TYLCV) infections [78]. 

### 3.3. Biotic and Abiotic Stress Transcriptomic Profiles Comparison

In this study, we analyzed and compared the transcriptomic response of R and S tomato plants exposed to fungi, bacteria, TSWV and *T. absoluta* as biotic factors and drought, salinity, low temperatures and oxidative stress as abiotic factors. A total of sixty-seven shared DEGs among different stresses, involved in cell wall metabolism, signaling TFs, and hormone pathways were found. 

The cell wall is a structure with sophisticated composition and organization, which interacts dynamically with sub-localized components. Perturbations in cell wall composition may induce different plant stress responses [79]. FLAs, a class of genes involved in cellulose deposition, were mainly activated by R plants to TSWV and during oxidative stress. By contrast, UGEs participating in pectin metabolism were induced by R genotypes to fungi and during low temperatures conditions. The R genotypes to fungi also activated GTs, which utilize nucleotide sugars as donor substrates to generate cell wall polysaccharides, and their implication in fungi resistance has been reported in different plant species [80,81].

A multitude of signaling events occurring downstream of the initial stress perception requires several RLKs, MAPKs and genes involved in calcium signaling. A list of signaling-responsive genes to both biotic and abiotic stress was obtained in our study. Among these, various RLKs (Solyc03g006100, Solyc11g017280 and Solyc02g091840) were found involved in response to *Phytophthora infestans* and *Colletotrichum coccodes* [82] or indicated as potentially involved in drought and pathogens recognition [46,83,84]. Interestingly, an RLK (Solyc03g078360) was activated by the R genotype to *R. solanacerarum* and resulted repressed in S plants inoculated with *Xanthomonas perforans* [85]. Fluctuations in calcium concentration can lead to phosphorylation events that promote plant stress response. Here, we found Solyc03g113390, a calcium-dependent protein kinase (CDPK), induced during abiotic stress and in response against different pathogens that may lead to specific defensive responses [86,87]. Moreover, the *Verticillium* wilt disease resistance Ve gene (Solyc09g005080) was found DE during different biotic stresses and in response to oxidative stress. It is worth noting that this gene was also found to be upregulated in a resistant line to TYLCV [88].

A combination of biotic and abiotic stress can trigger specific plant hormonal pathways. In this work, thirteen genes encoding for phytohormones and responsive to both biotic and abiotic stress were pointed out. Auxin-related genes, such as Small Auxin Up RNAs (SAURs), were activated during abiotic stress and by R genotypes to fungi, and four of them were repressed by the R genotype to TSWV. The induction of auxin-related genes can promote plant defense against abiotic stress and fungal pathogens by interacting with other phytohormones [89]. We also noted that ET (Solyc09g089580 andSolyc01g095080) was mainly activated during abiotic stress and fungi R plants and repressed by the TSWV R genotype. Furthermore, we found two genes involved in JA signaling that were downregulated during salinity (Solyc01g109140) and low temperature (Solyc07g054580) and induced by the R genotypes to TSWV and fungi. These results confirm that JA acts antagonistically with ET and auxin and its upregulation under TSWV infections could be particularly important in promoting plant resistance [90]. Finally, ABA (Solyc07g056570) was activated during different biotic and abiotic stress, supporting its participation in multiple stress responses [91,92].

Transcriptomic changes promoting defense against multiple stress are mediated by different TFs classes. The MYBs family was the most responsive family in our work since it is involved in the crosstalk response among stresses [93]. Interestingly, Solyc06g083900 (*SlMYB13*) was induced during salinity and repressed during low temperatures. This gene promotes tolerance to 2,4,6-trichlorophenol [94] and is activated in response to different pathogens [95,96]. Among TFs responsive to multiple stress, we also found a bZIP (*NPR1*; Solyc10g080770) involved in the regulation of PR gene expression [97]. It was induced by all the abiotic stress and by the R genotypes to fungi, while it was repressed by the R genotype to *T. absoluta*. The Solyc03g116890 (*WRKY39*) was involved in response to fungi, *T. absoluta*, TSWV, salinity and oxidative stresses. This gene is induced under heat and H_2_O_2_ stresses, and its overexpression induces *R. solanacerarum* resistance in cotton [98,99]. ERFs regulators control the expression of ET-dependent genes, which are well known to be involved in biotic and abiotic stress responses [100]. In this study, Solyc08g078180 (*SlERFa1*) was upregulated by R genotypes to fungi and bacteria and by oxidative stress, while it was downregulated in the TSWV R genotype. The Solyc08g078180, involved in cell death and signaling pathways against different pathogens, is induced by bacterial pathogens [101]. Moreover, Solyc11g072600, an AP2/ERF gene (*APETALA2d*), showed an opposite regulation during salinity and *T. absoluta* (upregulated) and low temperatures and R genotypes to bacteria (downregulated). Intriguingly, overexpression of a microRNA, conferring resistance against *Phytophthora infestans* infection, caused the downregulation of the Solyc11g072600 [102].

## 4. Materials and Methods

### 4.1. Bibliographic Research, Studys Selection and Transcriptomic Dataset Downloadg

A large-scale literature search was performed to find publicly available tomato (*Solanum lycopersicum*) RNA-seq experiments. Dozens of papers evaluating tomato stress response were collected. However, a further screening was made using the following criteria: (i) tomato plants were subjected to biotic or abiotic stress; (ii) sequencing was performed with Illumina technology; (iii) at least three biological replicates for treatment were used (Appendix A) except for the *T. absoluta* [29] and drought [22] experiments. Information on tomato transcriptomic studies of tomatoes exposed to biotic stress were mainly retrieved by [19]. In addition, three studies of tomatoes exposed to four different abiotic stresses were selected. In total, selected datasets comprised seven biotic stresses and five abiotic stresses (Table 1). Biotic stressors challenging tomatoes were: *Cladosporium fulvum* infecting tomato at 7 and 20 dpi (days post-infection), *Phytophthora infestans* at 40 dpi, *Pseudomonas syringae*, and *Ralstonia solanacearum* at 1 and 2 dpi, *Sclerotinia sclerotiorum* at 30 dpi, Tomato spotted wilt virus (TSWV) at 4, 7, 14, 21 and 35 dpi, and *Tuta absoluta* at 40 dpi. Transcriptomic studies of plants treated with *C. fulvum*, *P. infestans*, *R. solanacerarum*, TSWV, *T. absoluta*, and drought allowed the exploration of resistant (R), tolerant (T), and susceptible (S) responses. Response to abiotic stress was assessed by analyzing studies of tomato plants treated with drought (two different studies), salt, low temperature and oxidative stress (Table 1).

SRA accession number was used to access and download the corresponding raw sequencing data (fastq file) from the NCBI repository (Sequence Read Archive) (https://www.ncbi.nlm.nih.gov/sra, accessed on 9 September 2020 [103,104,105] using SRA Toolkit (2.10.8) (https://github.com/ncbi/sra-tools/wiki/01.-Downloading-SRA-Toolkit, accessed on 9 September 2020). After raw data collection, samples were grouped for experiments and comparative conditions.

### 4.2. Data Processing and Differential Expression Analysis

Collected raw data, including both pair-and and single-end reads, were analyzed with a standard pipeline through the online bioinformatics software AIR (Sequentia Biotech, Barcelona, Spain) using the RNA-seq package (https://transcriptomics.sequentiabiotech.com/, accessed on 11 January 2021). The analysis and comparison of RNA samples were conducted through algorithms described by [106]. After a quality check, which included the adapter removal and the trimming of low-quality reads, the clean reads were mapped against the reference tomato genome (version SL3.0). The number of reads before and after the quality check, the mean GC content, and the sequence length are reported in Appendix A. FeatureCounts was then used to obtain gene expression values (raw reads counts) [107]. The identification of differentially expressed genes (DEGs) was performed using the DESeq2 package [108]. Genes with a false discovery rate (FDR) < 0.05, using the Benjamini–Hochberg method, were considered differentially expressed (DE) and used for further analysis. DEGs with positive or negative logFC values were classified as upregulated (logFC > 0) or downregulated (logFC < 0), respectively. This workflow allowed us to compare and analyze different tomato studies starting from transcriptomic raw data.

### 4.3. DEGs Filtering Criteria

After AIR analysis, DEGs from biotic and abiotic experiments were downloaded and grouped. Then, comparisons among different RNA-seq studies were made by searching for common DEGs among datasets using the Microsoft Excel ^®^2013 (Ver. 15.0.5519) software.

### 4.4. Pathway Analysis

The differentially expressed genes were mapped to pathways using MapMan software (version 3.6.0, http://mapman.gabipd.org, accessed on 19 June 2022) [109]. In order to assign MapMan ontology to DEGs, we first downloaded the tomato protein annotation file in fasta format (ITAG3.0_proteins.fasta) from the Solgenomics database (https://solgenomics.net/ftp/tomato_genome/annotation/ITAG3.0_release/, accessed on 18 May 2022). Then, we used the Mercator tool (version 3.6) to create an ad hoc protein annotation file [110]. The Mercator file was then uploaded in MapMan version 3.6 and used as a mapping file together with the list of DEG considered for each comparison analyzed in this study.

## 5. Conclusions

In this study, a comparative analysis of twelve RNA-sequencing experiments of tomatoes exposed to biotic and abiotic stress was carried out to identify genes involved in defensive response against different stressors. Tomato response to biotic factors, such as *C. fulvum*, *P. infestans*, and *S. sclerotiorum* (fungi), *P. syringae*, and *R. solanacearum* (bacteria), Tomato spotted wilt virus (TSWV), and *T. absoluta*, was investigated. In addition, tomato challenged by abiotic stress (drought, salinity, low temperature, and oxidative stress) was also analyzed. This study allowed the identification of commonresponsive genes encoding for signaling proteins, cell wall precursors, TFs, and hormones, involved in response to biotic and abiotic stress. In particular, we analyzed in detail sixty-seven genes associated with the response to at least four different stresses. Among these, we found ten genes involved in the biosynthesis of cell wall compounds, thirty-six DEGs involved in signaling, thirteen DEGs participants in hormone biosynthesis, and eight DEGs encoding for TFs. Above all, we found different RLKs, MAPKs, Fasciclin-like arabinogalactans (FLAs), two glycosyltransferases, ten genes involved in the auxin., ET, and JA pathways, three MYBs, two bZIP, a WRKY, and an ERF gene. Our study provides a list of genes involved in response to multiple biotic and abiotic stress that could be tested in genetic engineering programs to improve tomato multiple-stress resistance.

## Figures and Tables

**Figure 1 ijms-24-04061-f001:**
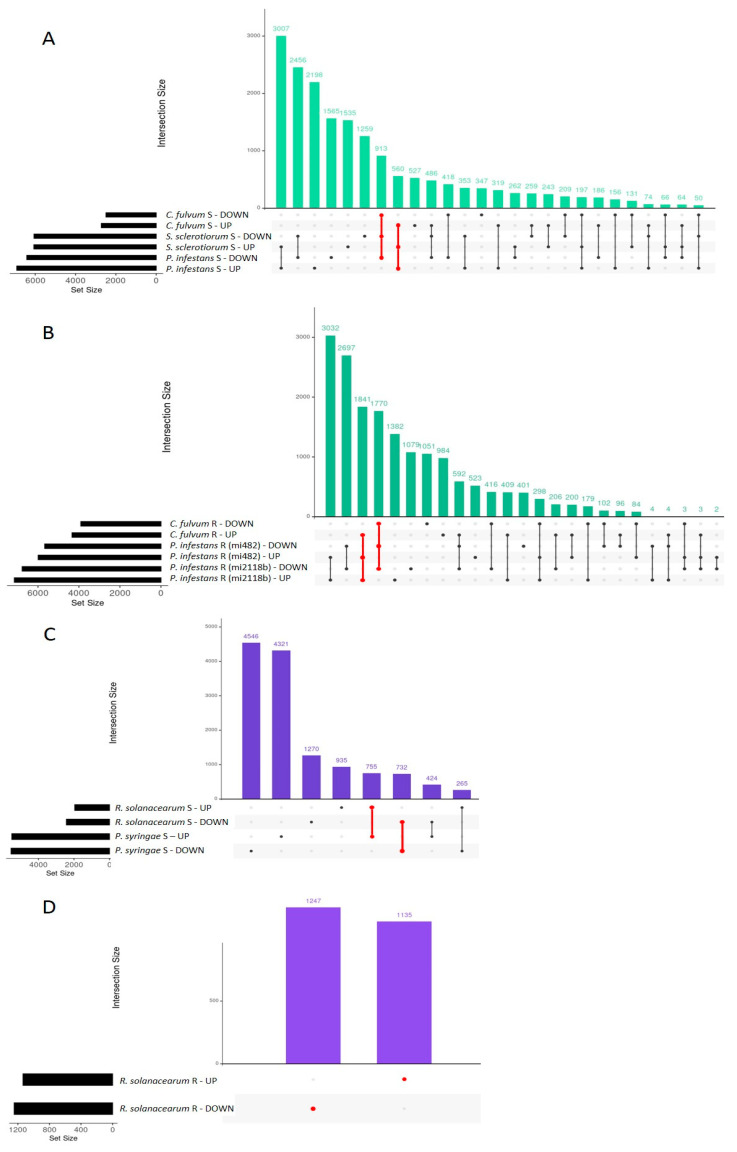
The intersection of up and down DEGs induced by tomato genotypes infected with fungi and bacteria (first 30 intersections). (**A**) Susceptible genotypes infected with fungi. (**B**) Resistant genotypes infected with fungi. (**C**) Susceptible genotypes infected with bacteria. (**D**) Resistant genotype infected with bacteria. In red are DEGs with the same expression trend in all the analyzed datasets.

**Figure 2 ijms-24-04061-f002:**
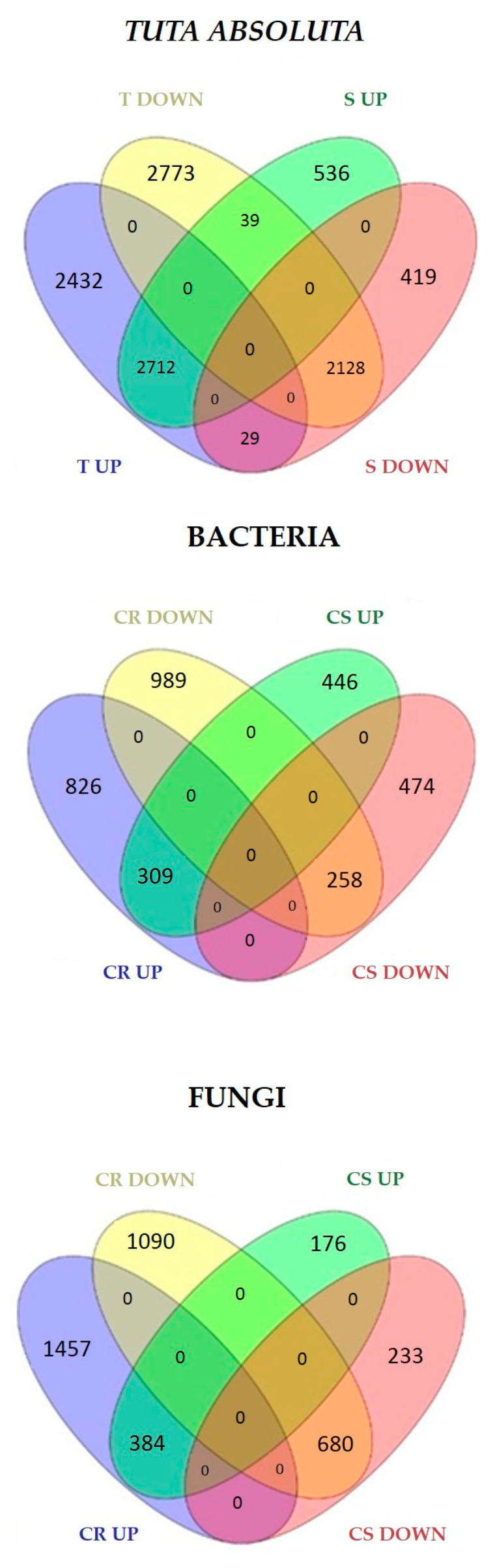
Private DEGs identified in R and S genotypes to fungi, bacteria, and *Tuta absoluta*. CR = common DEGs among R genotypes; CS = common DEGs among S genotypes; T = tolerant; S = susceptible genotypes.

**Figure 3 ijms-24-04061-f003:**
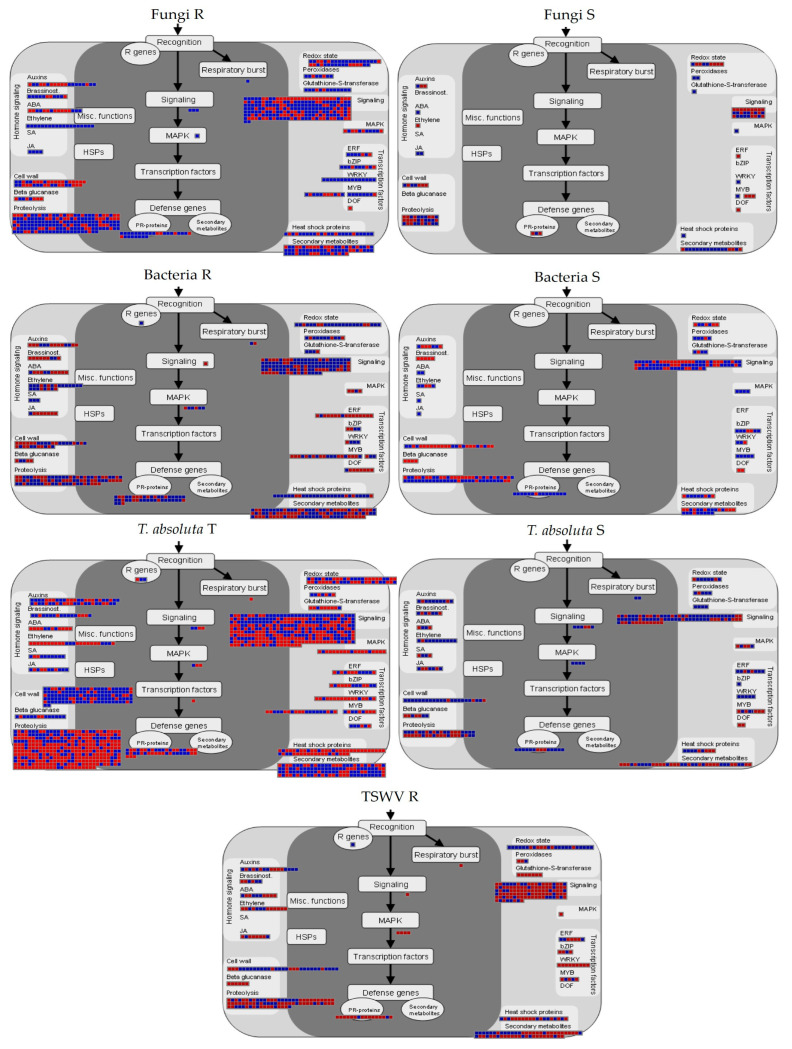
Genes putatively involved in biotic stress response under fungi, bacteria, TSWV, and *T. absoluta* infections. Blue: upregulated, red: downregulated genes. R = resistant; S = susceptible; T = tolerant genotype.

**Figure 4 ijms-24-04061-f004:**
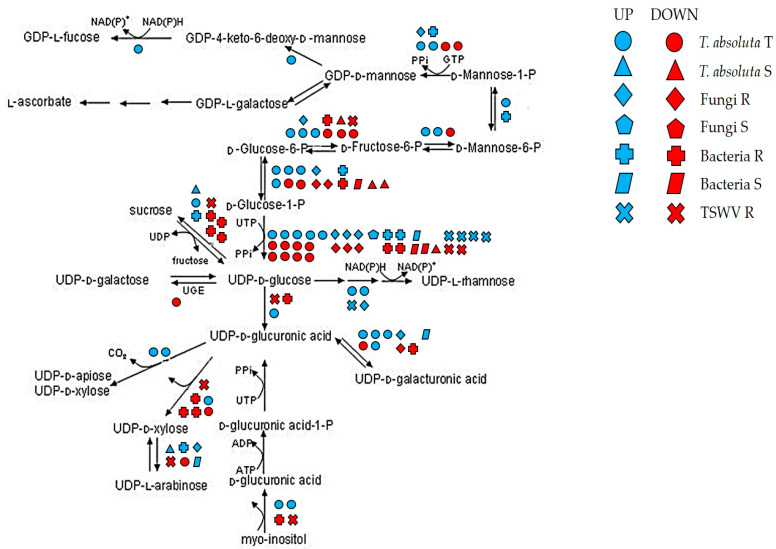
DEGs under various biotic stress involved in the biosynthesis of cell wall precursors. Blue: upregulated; red: downregulated. R = resistant; S = susceptible; T = tolerant genotype.

**Figure 5 ijms-24-04061-f005:**
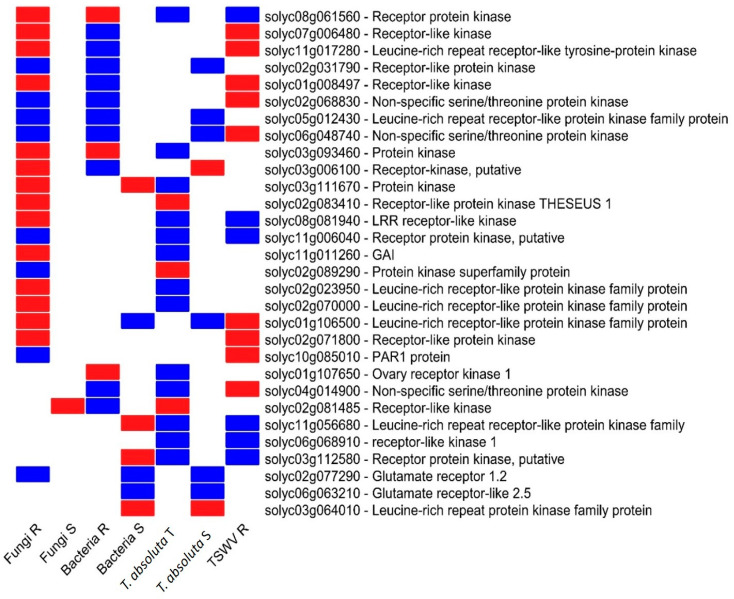
Heat map of DEGs participants in the signaling process. The expression pattern of DEGs identified and their function have been reported. Blue = upregulated, and red = downregulated genes. R = resistant; S = susceptible; T = tolerant genotype.

**Figure 6 ijms-24-04061-f006:**
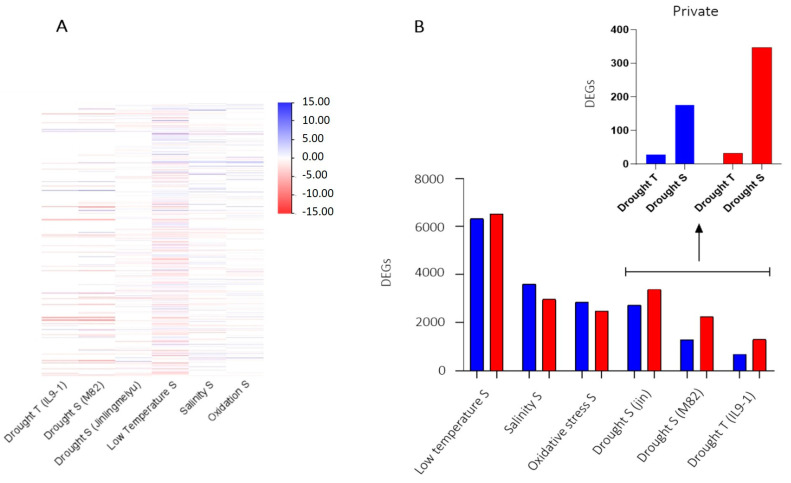
(**A**) Heatmap of DEGs induced during different abiotic stress; (**B**) Number of DEGs induced by different abiotic stress with a focus on private DEGs induced by tolerant and susceptible genotypes to drought. Blue = Upregulated and Red = Downregulated. R = resistant; S = susceptible; T = tolerant genotype.

**Figure 7 ijms-24-04061-f007:**
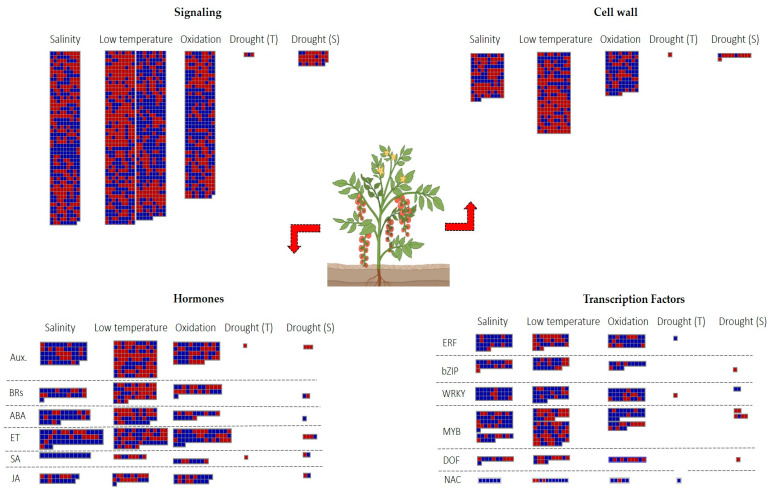
DEGs involved in signaling, cell wall, hormones, and TFs. Blue = upregulated; red = downregulated genes.

**Figure 8 ijms-24-04061-f008:**
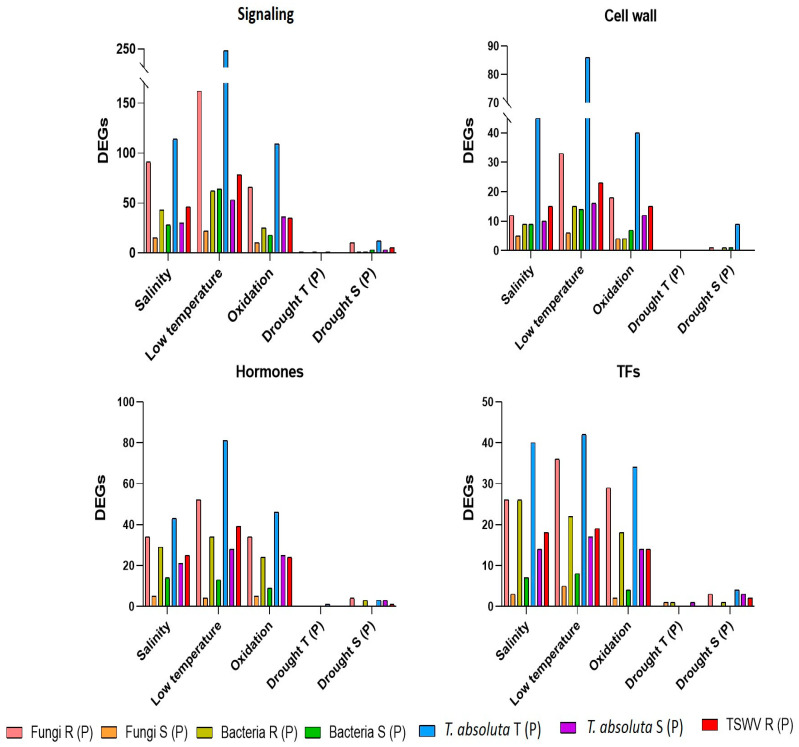
Common DEGs in biotic and abiotic stress related to signaling, cell wall, hormones and transcription factors (TFs). (P) = private DEGs; R = resistant genotype; S = susceptible genotypes; T = tolerant genotype.

**Figure 9 ijms-24-04061-f009:**
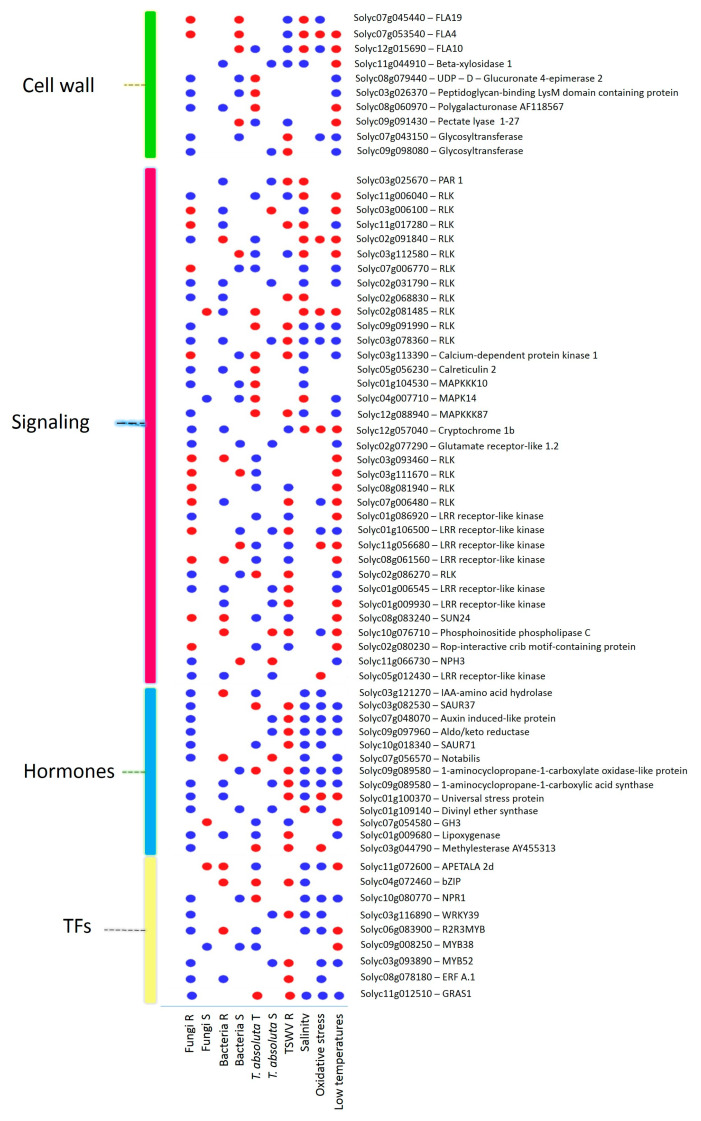
DEGs responsive to biotic and abiotic stresses, encoding for cell wall precursors, signaling proteins, hormones, and TFs. Blue = upregulated; red = downregulated genes. R = resistant; S = susceptible; T = tolerant genotypes.

**Table 1 ijms-24-04061-t001:** Transcriptomic responses to biotic and abiotic stress analyzed in this study. The number of up and downregulated genes and total differentially expressed genes (DEGs) has been reported for each experiment. Si = susceptible stressed; Ri = resistant stressed; Sni = susceptible not stressed; Rni = resistant not stressed; Ti = tolerant stressed; Tni = tolerant not stressed; Dpi = days post-induced stress.

Stress Type	Genotype		Comparison	Dpi	Up	Down	Total DEGs	SRA Code	References
	Resistant (R)	Susceptible (S)							
Biotic stress									
*C. fulvum*	CGN18423	MoneyMaker	Si. vs. Sni.	7 vs. 0	4698	4848	9546	SRP157120	[23]
			Si. vs. Sni.	20 vs. 0	2718	2487	5205		
			Ri. vs. Rni.	7 vs. 0	3562	3721	7283		
			Ri. vs. Rni.	20 vs. 0	4335	3904	8239		
*P. infestans*	Transgenic lines	M82	Ri_mi482_ vs. Rni _mi482_	40	5986	5667	11635	SRP168458	[24]
			Ri_mi2118_ vs. Rni_mi2128_		7150	6768	13918		
			Si. vs. Sni.		6914	6416	13330		
*P. syringae*	-	Ailsa Craig	Si. vs. mock	2	5500	5543	11043	SRP051074	[25]
			S. (treated) vs. mock		5684	5881	11565		
*R. solanacearum*	Hawaii 7996	West Virginia 700	Si. vs. Sni.	1 vs. 0	523	655	1178	SRP078159	[26]
			Si. vs. Sni.	2 vs. 0	1955	2426	4381		
			Ri. vs. Rni.	1 vs. 0	673	907	1580		
			Ri. vs. Rni.	2 vs. 0	1135	1247	2382		
*S. sclerotiorum*	-	Heinz	Si. vs. Sni	30	6065	6059	12124	SRP124841	[27]
Tomato spotted wilt virus (TSWV)	Fla8059.Sw7	Fla8059	Si. vs. Ri	4	18	76	94	SRP119544	[28]
			Si. vs. Ri	7	18	11	29		
			Si. vs. Ri	14	617	499	1116		
			Si. vs. Ri	21	722	768	1490		
			Si. vs. Ri	35	485	726	1211		
*Tuta absoluta*	BR221	PS650	Si. vs. Sni	40	3287	2576	5863	SRP286525	[29]
			Ti. vs. Tni		5176	4940	10116		
Abiotic stress									
Drought	IL9-1	M82	Si. vs. Sni	10	1283	2242	3525	SRP100604	[22]
			Ti. vs. Tni		682	1290	1972		
Drought	-	Jinlingmeiyu	Si. vs. Sni	5	2725	3374	6099	SRP156535	[20]
Salt	-	MicroTom	Si. vs. Sni	0.25	3600	2965	6565	SRP150651	[21]
Low temperature	-	Jinlingmeiyu	Si. vs. Sni	2	6323	6513	12836	SRP156535	[20]
Oxidation	-	MicroTom	Si. vs. Sni	0.25	2849	2475	5324	SRP150651	[21]

**Table 2 ijms-24-04061-t002:** Genes identified as responsive to different abiotic stress and involved in signaling and cell wall-related processes.

Gene ID	Function	Low T		Oxidation		Salinity		Drought S		Drought T	
		Up	Down	Up	Down	Up	Down	Up	Down	Up	Down
Signaling											
Solyc03g118040	Calnexin	X		X		X					X
Solyc02g072310	RLK		X		X		X				
Solyc05g056370	RLK		X		X		X				
Solyc06g063180	GLR2.2		X				X				
Solyc07g052390	GLR3.1		X				X				
Solyc07g052400	GLR3.2		X		X				X		
Solyc05g045650	GLR3.4		X		X						
Solyc02g067030	SNF1		X			X					
Cell wall metabolism											
Solyc01g009420	AXS	X			X		X				
Solyc02g067080	UGD	X			X		X				
Solyc07g006220	GAE	X			X		X				
Solyc05g050990	GAE		X				X		X		
Solyc08g080570	UGE		X			X		X			
Solyc03g097050	CSL	X		X		X		X			X

X = present as a DEG.

## Data Availability

The data presented in this study are openly available at Sequence Read Archive of NCBI repository (https://www.ncbi.nlm.nih.gov/sra, accessed on 9 September 2020) using the SRA codes reported in Table 1.

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
