# Peer review of "Comparison of Tomato Transcriptomic Profiles Reveals Overlapping Patterns in Abiotic and Biotic Stress Responses"

_ijms, 2023, doi:10.3390/ijms24044061_

Round 1

Reviewer 1 Report

To,

The Editor,

IJMS, MDPI,

Manuscript ID: ijms-2155711

Subject: Submission of comments of the manuscript in “IJMS"

Dear Editor IJMS, MDPI,

Thank you very much for the invitation to consider a potential reviewer for the manuscript (ID: ijms-2155711). My comments responses are furnished below as per each reviewer’s comments. 

In the reviewed manuscript, the authors analyzed and compared the transcriptomic responses of resistant and susceptible genotypes to seven biotic stresses (Cladosporium fulvum, Phytophthora infestans, Pseudomonas syringae, Ralstonia solanacearum, Sclerotinia sclerotiorum, tomato spotted wilt virus (TSWV), and Tuta absoluta) and five abiotic stresses (drought, salinity, low temperatures, and oxidative stress) to identify genes involved in response to multiple stressors. With this approach, we found genes encoding for TFs, phytohormones, or participating in signaling and cell wall metabolic processes, participating in defense against various biotic and abiotic stress. Moreover, a total of 1474 DEGs were commonly found between biotic and abiotic stress. Among these, 67 DEGs participated in response to at least four different stresses. In particular, we found RLKs, MAPKs, Fasciclin-like arabinogalactans (FLAs), glycosyltransferases, genes involved in the Aux., ET, and JA pathways, MYBs, bZIPs, WRKYs, and ERFs genes. Detected genes responsive to multiple stress might be further investigated with biotechnological approaches to effectively improve plant tolerance in field. The manuscript represents a very crucial information in a logical presentation. The study is well-conducted and provided important results that might use to improve multifactorial stress in plants. Therefore, it might be conditionally accepted subject to major revision. The authors need to address the following issues before it can be accepted for publication. 

  1. General note: the figures in this section are quite low resolution and difficult to make out. Higher-resolution versions will be needed for publication and text in the figure is not readable, for example, in Figures 1A, 1B, 1C, 1D, 2, 3, 7, and 9.
  2. Why you selected this crop for your experiment? Please provide the detail of the used variety.
  3. In Material and Methods:- indicate how many replicates assayed in each analysis/parameter. The number of samples or biological and technical replicates should be mentioned for each parameter in the methods.
  4. Material methods most the citation is the webpage, some website is not working, hence, better to cite the original research paper.
  5. Why authors not verified the expression data with qRT-PCR?
  6. The discussion should be interpreted with the results as well as discussed in relation to the present literature and authors must cite recently published research articles in the introduction and discussion on abiotic and biotic stress, for instance, "https://www.ncbi.nlm.nih.gov/pmc/articles/PMC8880425/

https://www.frontiersin.org/articles/10.3389/fpls.2021.663118/full

https://pubmed.ncbi.nlm.nih.gov/33673010

https://www.ncbi.nlm.nih.gov/pmc/articles/PMC8395832

https://bmcplantbiol.biomedcentral.com/articles/10.1186/s12870-020-02576-0

https://www.frontiersin.org/articles/10.3389/fpls.2021.748146/ful

  1. The authors must add a conclusion section.
  2. References: shall have to correct the whole References according to the ”Instructions for the Authors”, e.g. the Journal name and scientific name must be in italics.

Author Response

ANSWERS

We would like to thank the reviewer for providing important suggestions that allowed us to increase the quality of this manuscript. Moreover, minor text adjustments have been made to improve text clarity. Here below, we provided the answers to the reviewer's comments.

  1. General note: the figures in this section are quite low resolution and difficult to make out. Higher-resolution versions will be needed for publication and text in the figure is not readable, for example, in Figures 1A, 1B, 1C, 1D, 2, 3, 7, and 9.

We agree with the reviewer's comment. Hence, we provided new figures with increased quality and resolution. Moreover, Figure 4 has been modified to enhance clarity and readability. Finally, the figures included in the main text have also been provided as separate files to ensure their proper visualization.

  1. Why you selected this crop for your experiment? Please provide the detail of the used variety.

Tomato (Solanum lycopersicum) is one of the most extensively cultivated crops worldwide due to its nutritional quality and economic importance (Frusciante et al., 2007). Tomato is also used as a model system for plant genetic studies, and it is the most intensively investigated Solanaceous species (Barone et al., 2007). Moreover, the high-quality, well-annotated tomato reference genome makes it widely used to investigate the genetic basis of plants' stress interaction, and a large number of studies utilized this crop to explore the transcriptomic response to biotic or abiotic stress (Andolfo et al., 2021; Safavi-Rizi et al., 2020; Zhu et al., 2018; Zhao et al., 2020). In our study, we investigated and compared the transcriptomic datasets of different tomato varieties that had undergone biotic or abiotic stress. In most cases, the authors that performed the first transcriptomic study used varieties resistant or susceptible to specific stress. For this reason, in our study, we analyzed transcriptomic responses resulting from different varieties. In particular, used varieties were MoneyMaker, M82, Ailsa Craig, Hawaii 7996, West Virginia 700, Heinz, Fla8059, BR221, PS650, IL9-1, Jinlingmeiyu, and MicroTom. A comprehensive list of varieties has been reported in Table 1, along with the induced stress and their respective resistance or susceptibility to specific stress.

References:

Frusciante et al., 2007 - https://doi.org/10.1002/mnfr.200600158

Barone et al., 2007 - doi:10.1155/2008/820274

Andolfo et al., 2021 - https://doi.org/10.3390/genes12020184

Safavi-Rizi et al., 2020 - https://doi.org/10.1038/s41598-020-57884-0

Zhu et al, 2018 - https://doi.org/10.1016/j.cell.2017.12.019

Zhao et al., 2020 - https://doi.org/10.3389/fgene.2020.00540

  1. In Material and Methods: indicate how many replicates assayed in each analysis/parameter. The number of samples or biological and technical replicates should be mentioned for each parameter in the methods.

In Supplementary Materials, in Table S1, we added information about the number of biological replicates used for each analysis and condition (stressed, not stressed, and days post-infection). Moreover, we inserted an additional column with the name of the variety used in each treatment and a column with the codes to identify each biological replicate. In Materials and Methods at L. 516-517, we reported that " at least three biological replicates for treatment were used (Tab. S1) except for the T. absoluta [29] and drought [22] experiments."

  1. Material methods most the citation is the webpage, some website is not working, hence, better to cite the original research paper

We added citations to the corresponding research papers for the NBCI repository. For SRA Toolkit, no original research paper is associated. Hence, we provided a functional link to the dedicated webpage. We also checked the functionality of all the links provided in the manuscript.

  1. Why authors not verified the expression data with qRT-PCR?

Through a de novo analysis and a comparison of transcriptomic data retrivied from many studies focused on biotic and abiotic stresses response, we aimed to provide a list of candidate genes involved in response to different stress. Hence, our study represents a starting point for further functional studies on identified genes. For example, further studies might be conducted utilizing molecular and biotechnological approaches, and the expression of the most interesting genes can be verified under different conditions. In addition, RNA-seq, differently from microarray, is much more reliable in identifying differentially expressed genes because it does not suffer of concerns about reproducibility and bias and is robust enough (Coenye, 2021). Furthermore, there are a number of studies, reported by Everaert et al. 2017, that have specifically reported the  high correlation exsisting between  with RNA-seq and qPCR results.

  • Coenye T. Do results obtained with RNA-sequencing require independent verification? Biofilm. 2021 Jan 13;3:100043.

  • Everaert C, et al. Benchmarking of RNA-sequencing analysis workflows using whole-transcriptome RT-qPCR expression data. Sci Rep 2017;7(1):1559.

  1. The discussion should be interpreted with the results as well as discussed in relation to the present literature and authors must cite recently published research articles in the introduction and discussion on abiotic and biotic stress, for instance

https://www.ncbi.nlm.nih.gov/pmc/articles/PMC8880425/

https://www.frontiersin.org/articles/10.3389/fpls.2021.663118/full

https://pubmed.ncbi.nlm.nih.gov/33673010

https://www.ncbi.nlm.nih.gov/pmc/articles/PMC8395832

https://bmcplantbiol.biomedcentral.com/articles/10.1186/s12870-020-02576-0

https://www.frontiersin.org/articles/10.3389/fpls.2021.748146/full

In this work, we cited the most recent papers on biotic and abiotic stress responses in tomato. However, we included the most relevant citations as recommended. In particular, citations were added in the Introduction section at L. 38 and at L. 41 and in the Discussion section at L. 343 and L.403.

  1. The authors must add a conclusion section.

The original paper submitted to this journal was provided with a conclusive section at L. 569 – 586 after the materials and method section. To further clarify this point, here we provide the text contained in the Conclusion section:

In this study, a comparative analysis of twelve RNA-sequencing experiments of tomatoes exposed to biotic and abiotic stress was carried out to identify genes involved in defensive response against different stressors. Tomato response to biotic factors, such as C. fulvum, P. infestans, and S. sclerotium (Fungi), P. syringae, and R. solanacearum (Bacteria), tomato spotted wilt virus (TSWV), and T. absoluta was investigated. In addition, tomato challenged by abiotic stress (drought, salinity, low temperature, and oxidative stress) was also analyzed. This study allowed the identification of common-responsive genes encoding for signaling proteins, cell wall precursors, TFs, and hormones, involved in response to biotic and abiotic stress. In particular, we analyzed in detail sixty-seven genes associated with the response to at least four different stresses. Among these, we found ten genes involved in the biosynthesis of cell wall compounds, thirty-six DEGs involved in signaling, thirteen DEGs participants in hormone biosynthesis, and eight DEGs encoding for TFs. Above all, we found different RLKs, MAPKs, Fasciclin-like arabinogalactans (FLAs), two glycosyltransferases, ten genes involved in the Aux., ET, and JA pathways; three MYBs, two bZIP, a WRKY, and an ERF gene. Our study provides a list of genes involved in response to multiple biotic and abiotic stress that could be tested in genetic engineering programs to improve tomato multiple-stress resistance(L. 569 – 586).

  1. References: shall have to correct the whole References according to the" Instructions for the Authors", e.g. the Journal name and scientific name must be in italics.

We thank the reviewer for pointing out this inconsistency with the journal instructions. Hence, in the revised version of the manuscript, we checked all the references, added missing information, and wrote in italics the journals' and the scientific names.

Reviewer 2 Report

From the title and abstract, it can be seen that previous studies mainly focused on single stresses. In this study, the authors identified relevant genes responding to 7 biological stresses and 5 abiotic stresses by re-analyzing the previous transcriptomic data. Although the novelty of the work is undoubtedly high and the manuscript is well-written, some modifications efforts are needed to improve the text.

Point-to-point comments:

1. The data in Table 1 are difficult to understand. For example, for drought treatment, samples with different treatment times were taken for each stress treatment. Why did some of them choose 5d data [17], while some choose 10d data [15] in Table 1?

2. Why transcriptomic changes of tomato infected with Fungi, TSWV and T. absoluta were investigated at more than twenty dpi, whereas those infected with bacteria were investigated at two dpi? Since the expression of many genes changed in the early stage of stress treatment.

3. By comparing the previous data, the authors selected the simultaneously DE in at least one abiotic stress and three groups of biotic stresses. However, expression verification is lacked, which is suggested to supplement.  

4. The pixels of all the pictures are low and the pictures are indistinct.

Author Response

ANSWERS

We would like to thank the reviewer for providing important suggestions that allowed us to increase the quality of this manuscript. Moreover, minor text adjustments have been made to improve text clarity. Here below, we provided the answers to the reviewer's comments.

  1. The data in Table 1 are difficult to understand. For example, for drought treatment, samples with different treatment times were taken for each stress treatment. Why did some of them choose 5d data [17], while some choose 10d data [15] in Table 1?

In this work, an extensive bibliographic research was performed to collect transcriptome data about abiotic and biotic stress in tomato. We tried to collect data with similar time points and a minimum number of replicates. However, in some cases, it was not possible because many studies did not publicly release their raw data or they did not use a minimum number of biological replicates. Therefore, these works were excluded, and we only considered published works that released raw data and that were consistent with the selective criteria we posed. In addition, although the two drought studies were retrieved at two different time points, through the comparison of the DEGs lists obtained by the de novo analysis of both experiments, we identified common DEGs to both experiments, independently from the time point analyzed. Moreover, to increase the clarity of Table 1,  in the last column, we specified the works from which we retrieved the raw data, and the column's name has now been modified in Reference - Raw data.

  1. Why transcriptomic changes of tomato infected with Fungi, TSWV and T. absoluta were investigated at more than twenty dpi, whereas those infected with bacteria were investigated at two dpi? Since the expression of many genes changed in the early stage of stress treatment.

In this work, we compared the tomato transcriptomic response to  several stress conditions that typically manifest during tomato cultivation.  Based on our literature search, not all the studies that we found were made at the same time points. In addition, only for 10 works was possible to download the raw data or valuable raw data  (see material and methods for selective criteria posed) . So, we could analyze the early plant response for some stress while others were investigated at a late response phase. However, some studies were investigated at different times, making it possible to compare DEGs obtained at similar time points (TSWV, Fungi, and T. absoluta). While DEGs obtained by plants' transcriptomic response to bacteria were investigated at two dpi. Despite these differences, with our study, we provided candidate genes involved in response to stress independently from the time of occurrence of the stress. After identifying a candidate gene of interest, subsequent studies could be conducted to explore different temporal points of view.

  1. By comparing the previous data, the authors selected the simultaneously DE in at least one abiotic stress and three groups of biotic stresses. However, expression verification is lacked, which is suggested to supplement.

Through a de novo analysis and a comparison of transcriptomic data from many studies about biotic and abiotic stresses, we aimed to provide a list of candidate genes involved in response to different stress. Hence, our study represents a starting point for further functional studies on identified genes. For example, further studies might be conducted utilizing molecular and biotechnological approaches, and the expression of the most interesting genes can be verified under different conditions. In addition, RNA-seq, differently from microarray, is much more reliable in identifying differentially expressed genes because it does not suffer of concerns about reproducibility and bias and is robust enough to not always require validation by qPCR and/or other approaches (Coenye, 2021). Furthermore, there are a number of studies, reported by Everaert et al. 2017, that have specifically addressed the correlation between results obtained with RNA-seq and qPCR.

  • Coenye T. Do results obtained with RNA-sequencing require independent verification? Biofilm. 2021 Jan 13;3:100043.

  • Everaert C, et al. Benchmarking of RNA-sequencing analysis workflows using whole-transcriptome RT-qPCR expression data. Sci Rep 2017;7(1):1559.

  1. The pixels of all the pictures are low and the pictures are indistinct.

We agree with the reviewer's comment. Hence, we provided new figures with increased quality and resolution. Moreover, Figure 4 has been modified to enhance clarity and readability. Finally, the figures included in the main text have also been provided as separate files to ensure their proper visualization.

Reviewer 3 Report

Well written article in good English, long and detailed with numerous references in the literature. Brief and general introduction.

The figures, tables and supplementary materials are excellent and illustrative.

The results and the discussion were good, described in a clear, detailed way and which manage to re-elaborate the data of the inherent literature obtaining significant results.

Valid but reduced materials and methods.

Overall interesting study for the possible practical implications in agriculture and genetic engineering.

Figure S1 is missing

Par. 2.1.2 results not well represented graphically, accessions not identifiable

Author Response

ANSWERS

  1. Figure S1 is missing

We thank the reviewer for pointing out this inconsistency. Supplementary Figure 1 has now been included in the Supplementary Materials, as suggested.

Figure S1.Heat map of DEGs induced and repressed during different biotic stress

  1. 2.1.2 results not well represented graphically, accessions not identifiable

To facilitate the readers' better comprehension of the data shown in paragraph 2.1.2 (Figure 4), we generated a new version of the figure 4 (reported below).

Round 2

Reviewer 1 Report

Dear Editor,

Thank you for providing the opportunity to review the revised manuscript. The manuscript is improved considerably after revision according to the reviewer's comment. Now this study is a suitable contribution to the IJMS. I recommend the manuscript for publication.

Thank you

With best regards

Reviewer 2 Report

The author has solved all my questions.